# Phenolic Profile and Bioactivity Changes of Lotus Seedpod and Litchi Pericarp Procyanidins: Effect of Probiotic Bacteria Biotransformation

**DOI:** 10.3390/antiox12111974

**Published:** 2023-11-07

**Authors:** Junren Wen, Yong Sui, Shuyi Li, Jianbin Shi, Sha Cai, Tian Xiong, Fang Cai, Lei Zhou, Shengnan Zhao, Xin Mei

**Affiliations:** 1Key Laboratory of Agro-Products Cold Chain Logistics, Ministry of Agriculture and Rural Affairs, Institute of Agro-Products Processing and Nuclear-Agricultural Technology, Agro-Product Processing Research Sub-Center of Hubei Innovation Center of Agriculture Science and Technology, Hubei Academy of Agricultural Science, Wuhan 430064, China; jasminewen@webmail.hzau.edu.cn (J.W.); shijianbin@hbaas.com (J.S.); caisha@hbaas.com (S.C.); xiongtian@hbaas.com (T.X.); hmshi90@hbaas.com (F.C.); zhouleidjj@163.com (L.Z.); 202221582016@smail.xtu.edu.cn (S.Z.); 2College of Food Science and Technology, Huazhong Agricultural University, Wuhan 430070, China; 3School of Modern Industry for Selenium Science and Engineering, National R&D Center for Se-Rich Agricultural Products Processing, Hubei Engineering Research Center for Deep Processing of Green Se-Rich Agricultural Products, Wuhan Polytechnic University, Wuhan 430023, China; shuyi.li@whpu.edu

**Keywords:** lactic acid bacteria, procyanidin, lotus seedpod, litchi, bioconversion, antioxidant

## Abstract

Theoretically, lactic acid bacteria (LABs) could degrade polyphenols into small molecular compounds. In this study, the biotransformation of lotus seedpod and litchi pericarp procyanidins by *Lactobacillus plantarum 90* (*Lp90*), *Streptococcus thermophilus 81* (*ST81*), *Lactobacillus rhamnosus HN001* (*HN001*), and *Pediococcus pentosus* 06 (*PP06*) were analysed. The growth curve results indicated that procyanidins did not significantly inhibit the proliferation of LABs. Ultra-high-performance liquid chromatography high-resolution mass spectrometry (UPLC–HRMS) revealed that procyanidin B2 and procyanidin B3 in lotus seedpod decreased by 62.85% and 25.45%, respectively, with *ST81* metabolised, while kaempferol and syringetin 3-O-glucoside content increased. Although bioconversion did not increase the inhibitory function of procyanidins against glycosylation end-products in vitro, the 2,2′-Azinobis-(3-ethylbenzthiazoline-6-sulphonate) free radical scavenging capacity and ferric reducing antioxidant power of litchi pericarp procyanidins increased by 157.34% and 6.8%, respectively, after *ST81* biotransformation. These findings may inspire further studies of biological metabolism of other polyphenols and their effects on biological activity.

## 1. Introduction

*Receptaculum nelumbinis* (lotus seedpod) is the mature container of rosettes and is usually generated in large quantities as a by-product of lotus seed processing [1]. As a fruit distributed in subtropical regions, the abundant amount of *litchi chinensis* (commonly known as lychee or litchi) pericarp is produced by the related industries each year but not properly utilised [2]. Studies have shown that these by-products have a variety of benefits for human health; in particular, the procyanidins extracted from lotus seedpods (LSPC) and litchi pericarp (LPPC) exhibit favorable potential probiotic effects [3,4]. These phenolic compounds typically have flavan 3-ol nuclei structures that can be classified into A and B types based on their hydroxylation patterns [5]. Recently, the phenolic profile of LSPC and LPPC have been identified, mainly composed of monomers or polymers of catechins and epicatechins [1,6]. A large number of in vitro studies have demonstrated that procyanidins were effective in preventing oxidative damage caused by reactive oxygen species (ROS) in cell models [7,8,9]. Some studies have shown that procyanidin polymers tend to be degraded in the intestinal digestive phase in the presence of exogenous enzyme systems of gut microorganisms, and their products (mainly catechins or epicatechins) usually possess greater biological activity [10,11]. These conclusions suggest that the biotransformation of bacteria was conducive to enhancing the potential probiotic effects of phenolic acids.

Lactic acid bacteria (LABs), such as *L. plantarum*, *S. thermophilus*, *L. rhamnosus*, and *P. pentosus* have already been widely used in the food industry [12] due to their benefits on human well-being [13,14]. Recently, the interaction between polyphenols and LABs was partly clarified. Tabasco R et al. reported that the galloyl-esterase, decarboxylase, and benzyl alcohol dehydrogenase in *L. plantarum* could result in polyphenol degradation into gallic acid and phenol [15]. Ricci et al. further found some strains of *L. rhamnosus* and *L. plantarum* led to an increase in polyphenol and anthocyanin content in fermentation products [11]. In recent years, several reports have highlighted that the content of phenolic compounds was significantly increased by applying *Lp90* and *HN001* to convert by-products of plant-based materials (i.e., apple juice and grape pomace) [16,17]. Moreover, *ST81* and *PP06*, as common industrial dairy fermenters, have exhibited profound probiotic efficacy in vitro [18,19,20]. To sum up, it was evident that the use of LABs in fermented by-products of plant-based food may capably improve their phenolic acid activity and simultaneously open up a new field of fermentation applications for LABs.

To clarify the biotransformation effects of LABs on the phenolic profiles and biological activities of LSPC and LPPC, the LSPC and LPPC effects on the growth of *Lp90*, *ST81*, *HN001*, and *PP06* were first determined, and then UPLC–HRMS were used to monitor the metabolite composition of procyanidins at different sampling points. Furthermore, INFOGEST 2.0 protocol was applied to simulate the gastrointestinal (GI) digestion in vitro, the 2,2′-Azinobis-(3-ethylbenzthiazoline-6-sulphonate) free radical scavenging capacity (ABTS), oxygen radical absorbance capacity (ORAC), and ferric reducing antioxidant power (FRAP) were used to evaluate the antioxidant activity of procyanidins metabolites in chyme at each digestion stage, while AGE inhibition activity was evaluated by fluorescent method. Those results may expand new applications of procyanidins in the field of fermented food industry and food processing safety, and inspire further studies of biological metabolism of other polyphenols and their effects on biological activity.

## 2. Materials and Methods

### 2.1. Chemicals and Materials

The *Lp90*, *ST81*, and *PP06* for biotransformation were obtained from Wecare-bio (Suzhou, China), and *HN001* (HOWARU^®^) was purchased from DUPONT (Wilmington, DE, USA). The AB-8 macroporous resin used in the purification of LSPC was obtained from Molsu science equipment (Shanghai, China), and the C18 SPE column (Supelclean™ Ultra) for LPPC purification was purchased from Merck (Steinheim, Germany). UPLC–HRMS procyanidin standards (−)-Catechin (C0567) and (−)-epicatechin (E1735) were obtained from Sigma-Aldrich (Steinheim, Germany), and rutin (HY-N0148) from Med Chem Express (Monmouth Junction, NJ, USA). For the determination of antioxidant capacity, the 2,2′-azino-bis(3-ethylbenzothiazoline-6-sulfonic acid) (ABTS) and Trolox were purchased from TCI chemicals (Tokyo, Japan) and the 2,4,6-Tris(2-pyridyl)-s-triazine (TPTZ), 2,2′-Azobis(2-methylpropionamidine) dihydrochloride (AAPH), and fluorescein sodium salt were provided by Sigma-Aldrich (Steinheim, Germany). Pepsin A (P00791) and trypsin (T4799) used in simulated digestion were purchased from Sigma-Aldrich.

### 2.2. Extraction of LSPC and LPPC

LSPC was obtained using the method described by Wu et al., with modification [21]. Briefly, the 1 kg crushed frozen lotus seedpods were mixed with 10 L distilled water in 1:10 (*w*/*v*) at 80 °C, then 0.1% (*v*/*v*) NaHCO_3_ was added and incubated for 1 h, followed by centrifugation at 4000× *g* for 10 min—this process was repeated twice. The supernatant, with ethanol removed, was purified by using an AB-8 macroporous resin column. Lastly, the LSPC eluent was enriched by liquid–liquid extraction with ethyl acetate and then lyophilised after the removal of the solvents by vacuum distillation.

LPPC extract method was modified from Yao et al. [22]. A total of 1 kg of frozen litchi pericarps were mixed with 10 L ethanol solution (70%, *w*/*v*) and incubated at 50 °C for 2 h—the procedure was repeated twice. The supernatant, with ethanol removed, was filtered and purified by C18 Solid Phase Extraction (SPE) columns. The eluent was further enriched by liquid–liquid extraction with ethyl acetate, then lyophilised after the removal of the solvents by vacuum distillation. All the samples were stored at −20 °C for further experiments.

### 2.3. Growth Experiments

*Lp90*, *ST81*, *HN001*, and *PP06* were activated in *Man Rogosa Sharpe* (MRS) broth (Oxoid, San Jose, CA, USA) at 37 °C for 48 h. Subsequently, the bacterial cultures were diluted to approximately 7 Log CFU/mL to obtain the working solution. Native MRS broth inoculated with 2% working solution was added with different concentrations (0, 0.25, 0.5, and 1 mg/mL) of LSPC and LPPC, respectively, then maintained at 37 °C for 48 h. The bacterial density was measured at 2 h intervals at 600 nm (OD600) using a 722 N spectrophotometer (Inesa, Shanghai, China). The Gompertz equation was used for modeling growth kinetics according to Tlais et al., with minor modification [23].
y = k + Aexp{−exp[(μ_max e)/A] × (λ − t) + 1}
where A is the difference in OD600 units between the logarithmic and stabilisation periods, μmax is the maximum growth rate (OD600 units h^−1^), λ is the duration of the lag period (hours) and t is time (hours).

### 2.4. Procyanidin Metabolite Analysis

MRS broth inoculated with four LABs was added with 0.5 mg/mL LSPC and LPPC and sampled at 8 h intervals during the subsequent 48 h incubation process. The metabolites were extracted with ethyl acetate and further analysed with UPLC–HRMS, the medium without procyanidins was used as blanks.

#### 2.4.1. Total Phenolic Content (TPC) and Total Flavonoid Content (TFC)

TPC of LSPC and LPPC metabolites after 16 h of fermentation was measured by the Folin–Ciocalteu method [24], and the result was expressed in mg of gallic acid equivalents per g (mg GAE/g). Meanwhile, the TFC was examined using the aluminium chloride colourimetric method described by Akther et al., and the result was expressed in mg of rutin equivalents per g (mg RE/g) [25].

#### 2.4.2. UPLC–HRMS Analysis

Chromatography was performed on the Ultimate 3000 Ultra Performance Liquid Chromatography system (Thermo, San Jose, CA, USA); the conditions were modified according to the method described by Yao et al. [22]. The solution was separated on a Shim-pack C18 column (150 × 4.6 mm, 5 µm, Shimadzu Co., Kyoto, Japan) connected with a guard column (2.1 × 5 mm, Shimadzu). Eluent A was an aqueous solution containing 0.4% (*v*/*v*) acetic acid and eluent B was pure acetonitrile. The following gradient elution was used in this study: 0.01~40 min (95~65% A); 40~45 min (65~50% A); 45~50 min (50~20% A); 50~55 min (20~95% A); and 55~60 min (95% A). The column temperature was 30 °C and the injection volume was 10 µL. The mass spectrometer Orbitrap (Thermo Scientific, San Jose, CA, USA) was equipped with a high-energy spark-induced breakdown ionization (HESI-II) source operating in both positive and negative ionization modes. The optimized HESI-II and capillary temperature were set at 300 °C and 280 °C, respectively. MS data acquisition was performed in full scan/dd-MS^2^ mode, range in *m*/*z* 100–1500. The characterized phenolic were quantified by their corresponding standards.

### 2.5. AGE Inhibition Capacity after GI Digestion In Vitro

Glycated bovine serum albumin (BSA) was prepared according to the method of Li et al. [26]. The simulated GI digestion was modified from INFOGEST protocols and Pinto et al. [27,28]. Briefly, 14 mg glycated BSA and 2 mL LSPC and LPPC metabolites were mixed with 4 mL simulated gastric fluid (SGF) and 0.6 mL pepsin (25,000 U/mL), and the pH value was adjusted to 2 with 1 M HCl. After incubated at 37 °C for 2 h, the pH of the gastric surfactant was adjusted to 7 with 1 M NaOH, then immediately mixed with 2 mL of simulated intestinal fluid (SIF) and 0.625 mL trypsin (800 U/mL). This phase remained for 2 h at 37 °C, then quenched in a boiling water bath. Inhibition of fluorescence AGEs was determined using the method described by Wu et al. [29].

### 2.6. Antioxidant Capacity of Digestive Metabolites

Free radical-scavenging activity (ABTS and ORAC) and reducing power (FRAP) were adopted to comprehensively assess the antioxidant activity of metabolites during simulated GI digestion. The result was expressed in µmol of Trolox equivalents per mL (µmol TE /mL).

#### 2.6.1. ABTS Assay

The free radical scavenging activity of LSPC and LPPC metabolites was partly measured using the ABTS assay previously described by Bhattacharyya et al. [30]. This antioxidant capacity of the samples was represented by measuring the remaining ABTS absorbance at 734 nm after incubation compared to the control, and calculated according to the Trolox solution (10~180 μmol) standard curve.

#### 2.6.2. ORAC Assay

The ORAC of the samples was determined according to Wang et al., with minor modifications [31]. Briefly, 25 µL of appropriately diluted samples were added to the microtiter plate and subsequently mixed with 150 µL of fluorescein sodium salt (1 µmol) and 25 µL of AAPH (250 mmol). Blank and standard wells were then replaced with 25 µL of phosphate buffer or Trolox solution (10–180 µmol), respectively. All reaction wells were incubated at 37 °C for 15 min and a reading was taken every 5 min for 80 min (Ex.485 nm, Em.538 nm) in a microplate reader (Spark, Tecan™, Männedorf, Switzerland). Results were calculated using the difference of area under the fluorescence burst curve between the blank and the sample.

#### 2.6.3. FRAP Assay

The FRAP assay was slightly modified from previous reports [32]. Briefly, FRAP solution was prepared by mixing 20 mM FeCl_3_, 300 mM sodium acetate and 10 mM TPTZ solution in a 10:1:1 (*v*/*v*/*v*) ratio, protected from light. A total of 20 μL of sample or Trolox (10~180 μmol) and 280 μL of FRAP were added to a 96-well plate, and then the absorbance was measured at 593 nm immediately after 10 min of incubation at 37 °C.

### 2.7. Statistical Analysis

The obtained data were analysed by IBM SPSS (Statistical Package for the Social Sciences) version 25.0. The significance of means was performed by the Tukey test, and a two-tailed Pearson’s correlation test was performed to determine the correlation between changes in procyanidin content and antioxidant activity. Origin Pro 2021 (OriginLab, Northampton, MA, USA) and Adobe Illustrator version 28.0 (Adobe, San Jose, CA, USA) were used for further processing or plotting.

## 3. Results

### 3.1. LAB Growth Analysis

The growth kinetic curves of *Lp90* (A), *ST81* (B), *HN001* (C), and *PP06* (D) in MRS with different contents of LSPC are presented in Figure 1. Overall, LSPC did not cause a significant difference in λ and μmax of LABs compared to the blank, but variable reduction in the colony density at 600 nm during the stabilization phase was observed. Precisely, the growth of LABs was slightly inhibited by 0.5 mg/mL LSPC, but it did not exhibit further inhibition with increasing LSPC concentrations from 0.5 to 1 mg/mL. Incubation with LPPC resulted in similar effects to LSPC on the growth of LABs (Figure 1A–D). Whereas, LSPC appeared to have a superior inhibitory effect on the growth of LABs than LPPC, especially prolonging the lag period of HN001 (Figure 1D).

### 3.2. Procyanidin Metabolite Analysis

#### 3.2.1. Identification of LSPC and LPPC

As shown in Figure 2A and Appendix A. Briefly, seven compounds were identified in LSPC, which could broadly be classified according to the structure differences as flavanols (mainly (+)-catechin; S2), flavonols (such as kaempferol 3-O-glucoside; S5, myricetin 3-O-glucoside; S6 and syringetin 3-O-glucoside; S7) and procyanidins (such as procyanidin B2; S4 and procyanidin B3; S1), which were consistent with previous reports [2,32]. The polyphenol composition of LPPC was similar to LSPC. As shown in Figure 2A and Appendix A, four compounds were identified, mainly (+)-catechin (P1), (−)-epicatechin (P2), A-type procyanidin trimer (P3), and procyanidin A2 (P4).

#### 3.2.2. Characterization of LSPC Metabolite Composition

The phenolic profiles of the LSPC metabolites of four LABs are shown in Figure 2C and Appendix A. All the metabolites had significantly higher concentrations (*p* < 0.05) of kaempferol 3-O-glucoside than blank at each sampling point. Specifically, individual strains had different contributions to this flavonol content. The strains that induced the highest increase in kaempferol 3-O-glucoside were *Lp90* and *HN001*, which reached 0.91 and 0.28 mg (mean value), respectively, at the 40-hour sampling point. Similarly, *HN001* induced a 71.43% increase in syringetin 3-O-glucoside content. The contents of catechins and their multimers decreased to varying degrees with different strains. After 40-h biotransformation of LSPC with *Lp90, ST81, HN001*, and *PP06*, the content of procyanidin B3 decreased by 31.84%, 44.40%, 25.45%, and 41.50%, respectively. The procyanidin B2 content followed a similar trend, where decline rates of 65.71%, 85.71%, 62.85%, and 100%, respectively were observed. As the parent of the above procyanidins, the catechin content decreased by 40.32%, 60.18%, 26.67%, and 47.78%, respectively. In summary, these LABs had different biotransform capacities for phenolic acids. For instance, *ST81* has highly utilisation rate of procyanidin B3, procyanidin B2, and catechins, but its metabolites lack biologically active phenolic compounds. In contrast, *Lp90* and *HN001* preferred to utilise procyanidin multimers, while the increased content of kaempferol 3-O-glucoside and syringetin 3-O-glucoside in metabolites were observed.

#### 3.2.3. Characterization of LPPC Metabolite Composition

The LPPC metabolites of LABs are shown in Figure 2D and Appendix A. Overall, the procyanidin content under the induction of LABs underwent different degrees of decline. High degradation rates of procyanidin A3 by LABs were observed in the study, with *Lp90*, *ST81*, *HN001*, and *PP06* contributing to decreases of 64.50%, 82.26%, 100.00%, and 56.45%, respectively, at 40 h sampling point. (−)-Epicatechin had a similar trend with declining rates of 61.87%, 64.07%, 82.01%, and 58.27%, respectively. In addition, catechins were undetectable after 36 h of incubation except for *HN001* metabolites. In addition, LABs did not tend to degrade the procyanidin A2 in LPPC, except *HN001*; the reduction rates of *Lp90, ST81, HN001*, and *PP06* were 35.48%, 59.14%, 29.03%, and 27.97%, respectively, at 40 h sampling point.

### 3.3. Inhibition Effect of LSPC and LPPC Metabolites on AGEs Fluorescent Formation

Figure 3 illustrates the ability of the LSPC and LPPC metabolites of LABs after 16 h incubation to inhibit fluorescent AGEs in simulated digested intestinal chyme. The metabolites at the 16 h sampling point were chosen for the assessment of biological activity because further biotransformation of LABs would result in a decrease in procyanidin content (Figure 2). The fluorescent AGE inhibitory capacity in LSPC metabolites of *HN001* was 39.17%, while that of LPPC metabolites of *Lp90* was 50.69%. However, natural LSPC and LPPC were seen to have better fluorescent AGE inhibitory capacity during simulated digestive, with a difference of 13.29% and 12.90%, respectively, compared to the above metabolites.

### 3.4. Antioxidant Capacity Effect of LSPC and LPPC Metabolites

#### 3.4.1. Antioxidant Capacity Analysis

FRAP, ABTS, and ORAC methods were used to comprehensively evaluate the antioxidant capacity of the LSPC and LPPC metabolites of LABs, and the results are presented in Figure 4. The metabolites’ antioxidant capacities of gastric chyme were remarkably superior (*p <* 0.05) than the intestinal chyme in a statically simulated digestive system. LSPC and LPPC metabolised by *Lp90*, *ST81*, *HN001*, and *PP06* exhibited enhanced scavenging of ABTS radicals, ferrous ions, and reactive oxygen species compared to the natural procyanidin (Figure 4B,E). Among them, the metabolites of HN001 demonstrated the greatest increment in antioxidant activity—the FRAP, ABTS, and ORAC increased by 256.29%, 53.27%, and 123.66%, respectively, in LSPC metabolites intestinal chyme, while in LPPC metabolites intestinal juice, the increases were 166.43%, 32.65%, and 82.65%, respectively. Notably, although the content of each procyanidin compound in LSPC and LPPC was reduced after metabolism by LABs compared to blank (Figure 2C,D), this did not seem to affect their antioxidant capacity. For instance, the content of catechin, epicatechin, procyanidin A3, and procyanidin A2 decreased by 65.22%, 35.25%, 58.06%, and 35.48%, respectively, in LPPC metabolites of *S. thermophilus* at 16 h sampling point, while the FRAP, ABTS, and ORAC of LPPC were higher than blank by 157.34%, 6.8%, and 10.17%, respectively.

#### 3.4.2. Correlation Analysis between Antioxidant Activity and Metabolite Composition

Overall, the enhanced TPC and TFC in metabolites resulting from the biotransformation of LABs were strongly correlated with antioxidant activity (Figure 5 and Appendix A). In terms of LSPC metabolite, glycosylated flavonoids provided an excellent antioxidant, i.e., the r^2^ between the net increase in S5 and FRAP, ABTS, and ORAC ranged from 0.7 to 0.9, with the correlation for FRAP and ORAC in the intestinal phase being more significant, with r^2^ of 0.903 and 0.947, respectively. Whereas, all procyanidin multimers showed different degrees of negative correlation with antioxidant activity. In LSPC, changes in S3 content had the most pronounced effect on FRAP in the intestinal phase, with an r^2^ of 0.974. In LPPC, the situation was different, Figure 5B shows that a decrease in P3 content had a positive effect on the enhancement of the antioxidant activity of the metabolites (r^2^ of 0.929). In addition, the increase in catechin content did not seem to have a significant effect on the overall antioxidant activity in either LSPC or LPPC (r^2^ was in the range of −0.3 to −0.4).

## 4. Discussion

Certain concentrations of phenolic acid could scavenge free radicals while maintaining excellent inhibitory capacity of LABs’ growth. Phenolic acid inhibits the growth of LABs mainly by disrupting their cell wall and membrane structure, further reducing microbial acid production and carbohydrate utilisation [33]. Whereas, Laosee et al. recently revealed that the growth of LABs can be promoted by polyphenolic-rich fruit juices [34]. Therefore, it is important to choose a suitable procyanidin concentration to observe the biotransformation model of LABs. Although 0.25 mg/mL of procyanidins did not significantly affect the growth of LABs (Figure 1), metabolites were not detectable at any sampling points in the pre-experiment of this study. While 0.5 mg/mL of LSPC or LPPC may slightly affect the growth rate of LABs, the total number of colonies was still higher than 8.0 Log CFU/mL during the stabilisation period, which did not have a negative impact on experimental results. Therefore, 0.5 mg/mL of procyanidins were used in the subsequent experiments.

To our knowledge, *L. plantarum* has the ability to biotransform catechins into smaller units. He et al. suggested that the endogenous enzyme system is the main contributor to the degradation of hydroxycinnamic acid and hydroxybenzoic acid [35], and it has been shown that *L. rhamnosus* is also able to convert protocatechuic acid [11]. Whereas, not all the LABs had the same trend in procyanidin conversion, which suggested the existence of multiple metabolic pathways [36]. Relevant studies found that glycosylated flavonoids were systematically depolymerised by LABs due to the β-glucosidase activity, leading flavonoid content to decline throughout the incubation process [37,38,39,40], while *Lp90* and *HN001* were able to enhance the content of glycosylated flavonoids in our research. One possible explanation was that some strains of LABs can express glycosyltransferases, which can be involved in the glycosylation of a wide range of polyphenols [41]. Xiao et al. and Zi et al. suggested that microbial glycosyltransferases typically link a 7-O-glycoside for the flavonoid and tend to provide a 3-O-glycoside if the flavonoid has a hydroxyl group at the C-3 position, the latter tends to enhance the antioxidant or antidiabetic potential of the product [42,43]. This pathway enables LABs to detoxify and solubilise phenolics in the cell membrane. The ability of the enzymes to glycosylate quercetin and catechols glycosylation has been reported [44]. Additionally, by purifying the expression products of ram1 (lp_3471) and ram2 (lp_3473) genes for α-rhamnoses from *L. plantarum* WCFS1 in *Escherichia coli*, Ferreira et al. examined their ability to hydrolyse some of the flavonoids and carbohydrate-related substrates and showed that in comparison to Ram1, Ram2 was only able to convert rutin to monoglycosylated quercetin-3-O-glucosides due to the lower β-glucosidase activity [45]. In conclusion, it could be speculated that the production of glycosylated flavonoids in the present study may be attributed to the fact that the selected LABs had lower β-glucosidase expression, thus exhibited a net increase in S5 content due to the specific kaempferol glycosyltransferases activity. However, a more targeted analysis of the mechanism of glycosylation by enzyme systems expressed by LABs is still required.

Recent reports have claimed that fermentation of green tea by *L. plantarum* resulted in a decrease in the content of catechins and their derivative polymorphs, and was accompanied by a significant increase in gallic acid [46]. However, the procyanidin content in this study decreased rapidly and stabilised with the incubation time extended to 16 h. Li et al. reported a continuous decrease in TPC after apple juice fermentation, which was attributed to the continuous catabolism of phenolic compounds by the enzyme system of LABs [47]. However, this phenomenon does not always mean that the bioactivity of the metabolites was reduced, Ru et al. found that biotransformation by probiotics resulted in a decrease in the reduction in the polyphenol component of walnut flowers [48], while an increase in the total polyphenol content (Folin–Ciocalteu method) of the metabolites was observed, which may result in lesser impact on the antioxidant capacity of the partial metabolites. Therefore, considering the presence of certain phenolic hydroxyl groups during metabolism that cannot be detected by chromatographic methods, TPC, TFC, and product bioactivities were analysed in subsequent studies.

In addition to the changes in the metabolite compositions of LSPC and LPPC, we further examined their biological activities. Previous studies have shown that catechins and epicatechins could provide superior inhibition of the formation of fluorescent AGEs than aminoguanidines [26,49]. In this study, the LSPC metabolite of *HN001* had higher levels of catechin and rutin content (Figure 2C and Appendix A), which partly explains its superior inhibition of fluorescent AGEs. Some studies indicated that glycosylation of flavonoids would tend to reduce AGE inhibitory capacity [50,51,52]; relatively high contents of the kaempferol 3-O-glucoside and syringetin 3-O-glucoside in LSPC metabolites did not significantly affect the AGE inhibitory capacity in this research. (−)-Epicatechin and its polymers were the main components of LPPC. Generally, polymers of procyanidin had poorer resistance against glycation compared to monomers [53]. The (−)-epicatechin content in *ST81* and *PP06* metabolites of LPPC was 0.84 and 0.90 mg (Figure 2D and Appendix A), respectively, which were significantly higher than other LAB metabolites of LPPC (*p* < 0.05) and resulted in a relatively high inhibitory capacity of the AGEs. The inhibitory effect of natural LSPC and LPPC on the fluorescent AGEs was more pronounced compared with metabolites. One of the reasons was the higher phenolic acid contents of LSPC and LPPC, which implies that more phenolic compounds can inhibit fluorescent AGEs during the intestinal digestion stage [54]. Another reason was the glycated bovine serum model used in this experiment, which binds more easily to untransformed phenolic acids [55].

In the antioxidant activity studies, we concluded two possible reasons for the reduced antioxidant activity of metabolites in the intestinal phase. Firstly, the antioxidant activity of phenolic acids could be inhibited by the neutral pH value in simulated intestinal fluid [56]. Secondly, parts of LSPC and LPPC have been complexed with glycated bovine serum to inhibit the release of AGEs during gastric digestion, partially reducing the amount of procyanidins to bonded free radicals [57]. The former was also responsible for the significantly higher antioxidant activity of metabolites in the gastric phase than intestinal phase. Additionally, the glycosylated flavonoids may have lower inhibitory activity of fluorescent AGEs than parents, but their excellent antioxidant activities have been reported [56], which resulted in relatively higher antioxidant activity in vitro of LSPC metabolites by *HN001*. The second conjecture was verified in subsequent correlation analyses. It is also known from the statistical analysis that kaempferol 3-O-glucoside was more inclined to scavenge ROS, which is attributed to its potentially strong hydrogen or electron-donating capacity [58]. Furthermore, the content of catechins and/or epicatechins did not have a positive correlation with the antioxidant activity of the metabolites observed in this study; although microbial biotransformation reduced the content of the above monomers, TPC, and TFC were significantly higher than those of the unfermented procyanidins (Appendix A), which could be explained by the hydrolysis of phenolic acids by microbial enzymes into smaller molecules that could not be detected under the chromatographic methods used in this study, whereas phenolic hydroxyls in these compounds tend to exhibit a higher degree of flexibility compared with the parent, which makes it easier to contact with free radicals and exhibit antioxidant activity [47]. In addition, the results of this study also pointed out that LABs that are capable of efficiently transforming procyanidin B2 and A-type procyanidin trimer typically result in metabolites with enhanced antioxidant activity, which could also be useful for identifying other microorganisms with potential biotransformation activities.

In summary, the study confirmed that the selected four LABs were able to bioconvert LSPC and LPPC at certain concentrations with favourable antioxidant activity and considerable AGE scavenging ability of the products. However, some limitations of this study must also be recognised. On the one hand, this study only describes the possible causes of flavonoid glycosylation by LABs, but does not determine the β-glycosidases or related glycosyltransferases that microorganisms may produce during biotransformation. On the other hand, static in vitro simulations of digestion have large differences from real in vivo behaviour, which may generate biases in biological activity, and thus affect the accuracy of the study results. To fully understand the effects of biotransformation on the phenolic profiles and biological activities of LSPC and LPPC, further studies should analyse the content of glycosyltransferases or glycosidases expressed by microorganisms in the presence of procyanidins or measured at the level of gene expression. Changes in the antioxidant activity of metabolites should also be explored by in vitro dynamic simulation of digestive systems or animal models. By conducting such studies, further mechanisms of LAB-mediated degradation of procyanidin multimers and synthesis of glycosylated flavonoids would be discovered, so as to identify appropriate LAB strains and fermentation methods.

## 5. Conclusions

In this study, 0.5 mg/mL LSPC or LPPC did not significantly affect the growth of the four LABs selected in this research. Partial procyanidin dimer or trimer in LPPC and LSPC were degraded into (+)-catechins or (−)-epicatechins. Moreover, the increase of kaempferol 3-O-glucoside, myricetin 3-O-glucoside, and syringetin 3-O-glucoside was observed in the LSPC metabolite phenolic profile. In a further simulated digestive system, the excellent in vitro antioxidant potential of LSPC and LPPC metabolites was of interest compared to natural procyanidins, despite the insignificant scavenging ability of fluorescent AGEs. The results suggested that the biotransformation of LABs enables the high-value utilisation of by-products from the litchi and/or lotus seedpod in the functional foods industry, to obtain metabolites with favourable antioxidant potentials, especially *HN001.* In addition, further clarification of the glycosylation formation mechanism of flavonoids in LSPC metabolites is conducive to optimising the scavenging activity of the metabolites for fluorescent AGEs, and thus expanding their applications.

## Figures and Tables

**Figure 1 antioxidants-12-01974-f001:**
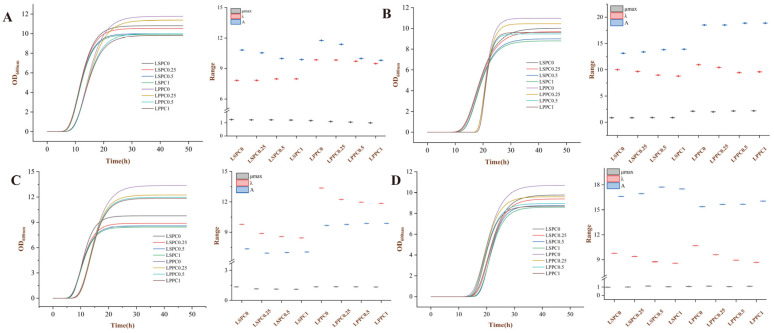
Effects of different concentrations of LSPC and LPPC on LAB growth. (**A**–**D**) Kinetics growth of *Lp90* (**A**), *ST81* (**B**), *HN001* (**C**), and *PP06* (**D**) in MRS broth with LSPC and LPPC at 37 °C.

**Figure 2 antioxidants-12-01974-f002:**
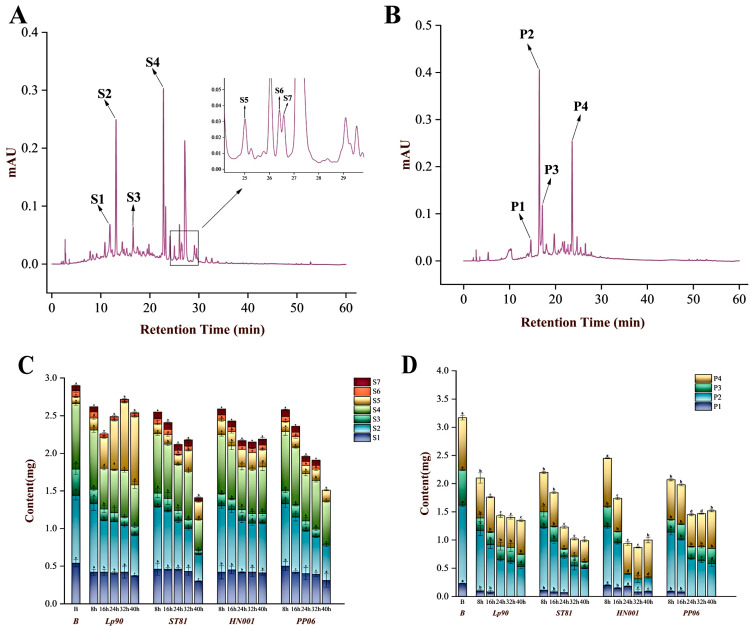
The original procyanidin and LAB metabolite compositions. Representative chromatograms of the identified phenolic compounds in LSPC (**A**) and LPPC (**B**). Phenolic compounds identified in LSPC: (S1) procyanidin B3; (S2) (+)-catechin; (S3) procyanidin B2; (S4) rutin; (S5) kaempferol 3-O-glucoside; (S6) myricetin 3-O-glucoside; and (S7) syringetin 3-O-glucoside. Phenolic compounds identified in LPPC: (P1) (+)-catechin; (P2) (−)-epicatechin; (P3) A-type procyanidin trimer; and (P4) procyanidin A2. Content of individual phenolic compounds in LSPC (**C**) and LPPC (**D**) at different sampling points (mg), and different lowercase letters indicate significant differences between sampling times by the Tukey test (*p* < 0.05).

**Figure 3 antioxidants-12-01974-f003:**
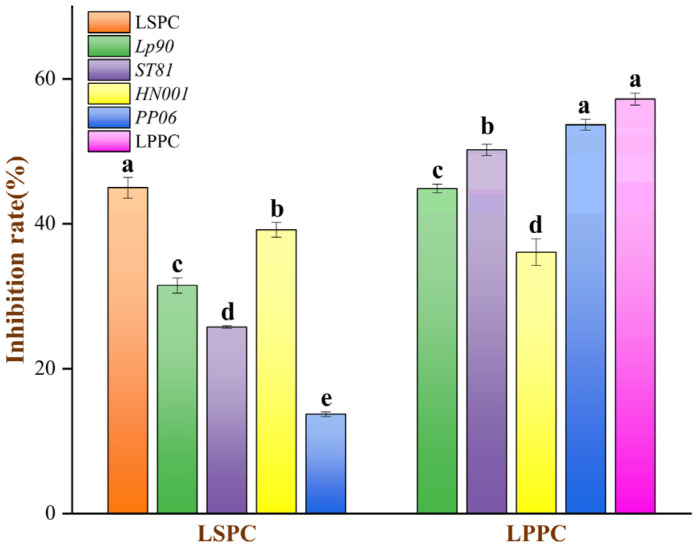
Inhibition rates of LSPC and LPPC metabolite on fluorescent advanced AGEs. Data shown are mean ± SEM, and different lowercase letters indicate significant differences between metabolites of various LABs by the Tukey test (*p* < 0.05).

**Figure 4 antioxidants-12-01974-f004:**
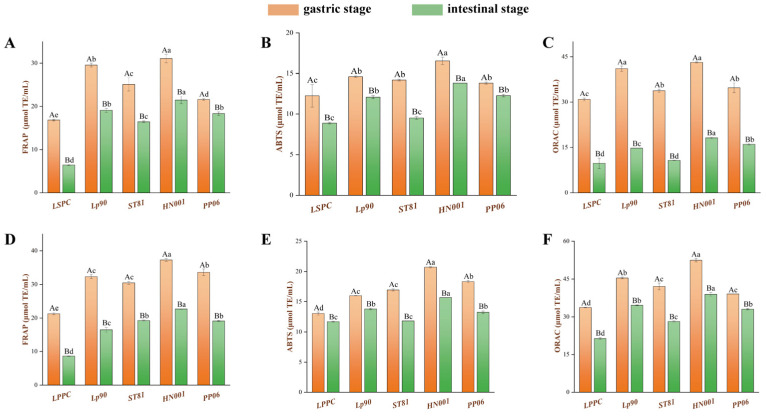
Antioxidant activity of metabolites. The FRAP, ABTS, and ORAC activity of LSPC (**A**–**C**) and LPPC (**D**–**F**) metabolites after gastrointestinal digestion in vitro. Data shown are mean ± SEM, and different uppercase letters indicate significant differences between gastric and intestinal chyme, while the lowercase letters indicate significant differences between metabolites of various LABs by the Turkey test (*p* < 0.05).

**Figure 5 antioxidants-12-01974-f005:**
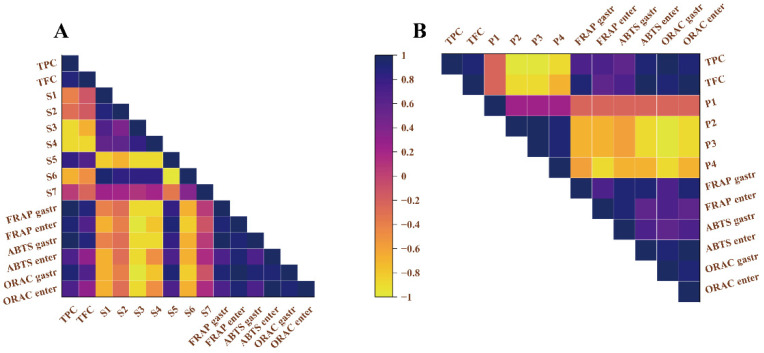
Correlation analysis of LSPC (**A**) and LPPC (**B**) metabolites with antioxidant activities determined by diverse assays in vitro.

## Data Availability

No new data were created or analyzed in this study.

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
