# Peer review of "Phenolic Profile and Bioactivity Changes of Lotus Seedpod and Litchi Pericarp Procyanidins: Effect of Probiotic Bacteria Biotransformation"

_antioxidants, 2023, doi:10.3390/antiox12111974_

Round 1
Reviewer 1 Report
Comments and Suggestions for Authors
The aim of the research included in the manuscript was to assess the biotransformation of procyanidins of lotus seed pods and litchi pericarp by selected lactic acid bacteria. The research included in the manuscript is innovative. The analytical methods used are standard and the references used are appropriate. The text of the article requires some additions, which are presented below.
On what basis was the selection of lactic acid bacteria used in the research made? Was the selection made on the basis of preliminary research or were other factors taken into account during the selection? The choice of L. plantarum and S. thermophilus is clear, as these are bacteria commonly involved in the fermentation of vegetables and dairy products. The justifying sentence should be included in the manuscript.
The description of LSPC and LPPC extraction lacks information on the amount of processed material. Lack of this information makes it impossible to repeat the experiment. How much lotus seedpod and frozen litchi pericarps were taken for extraction? How many times was the extraction procedure performed?
In the discussion, I recommend comparing the activity of β-glucosidase and glycosyltransferases of the tested strains of lactic acid bacteria published earlier. If such studies are not planned, in my opinion it is worth comparing with the activity studied by other researchers assessing the activities of enzymes of various strains.
In the description of analytical methods it is described that “The result was expressed in mg of Trolox equivalents per g (mg TE /g)” (L148) while in Figure 4 the results are given in µmol/mL. What is the reason for such discrepancies?
The captions on the charts in Figure 4 are very illegible. I recommend increasing the font size in the axis description and maybe making it more readable.
In the discussion of the results, I recommend comparing the obtained results of antioxidant capacity tested using different analytical methods. What may be the reason for differences between the results of the same tests, but determined using different methods?
The description of the statistical methods used states that “The significance of means was performed by Tukey test” while in the caption of chart 4 it is written that “…while lowercase letter indicates significant difference between LABs by the Dunnett test”. Information should be provided on what tests are used to assess significant differences between the results. Was there any justification for including a different test in the antioxidant capacity results? This should be mentioned in the description of the statistical methods, justifying the choice.
Reviewer 2 Report
Comments and Suggestions for Authors
In the present study, the authors have analyzed the effect of biotransformation of lotus seedpod and litchi pericarp procyanidins into polyphenols with small molecular size mediated by LABs.
Although there are some typographical errors in the submitted manuscript, the aim of the present study seems to be of interest and the research seems to be well designed and conducted. However, the descriptions in the discussion section seems to be insufficient for satisfying the criteria for publication, so, there are some concerns in the manuscript for acceptance at present form as follows:
Major
1. How did the authors select LAB strains? As you know, the characteristics of LABs have reported to be strain specific rather than strain, thus differences on between not only the selected strain but also the other strains of those are important information.
2. The authors analyzed and compared the biotransformation mediated by some LAB strains, however, it was not enough discussed what kinds of differences were affected to the results. Thus, the authors should add those description based on genetic characteristics and metabolic profiles with inter- and intra-species comparison.
Minor
There are some typographical errors in the manuscript.
e.g. Genus name should be written in italics, and the name also should be spelled out on the first use.
Comments on the Quality of English LanguageThere are some typographical errors in the manuscript.
e.g. Genus name should be written in italics, and the name also should be spelled out on the first use.
Reviewer 3 Report
Comments and Suggestions for Authors
In my opinion, the manuscript entitled Phenolic profile and bioactivity changes of lotus seedpod and litchi pericarp procyanidins: Effect of probiotic bacteria bio-transformation has several issues, such as:
1. Line 21 – are instead of were. It is a theoretical affirmation, and an abstract does not begin with an affirmation or an obtained result.
2. Line 25. Lactobacillus must be written in italic.
3. UPLC-HRMS please mentioned under brackets the meaning of the used abbreviation in order to be clear for the readers.
4. Line 26. S. thermophilus should be written in italic.
5. Line 27. I think that bioconversion instead of bioconvert.
6. In the introduction part, authors should mention some ideas about Lotus seedpods (LSPC) and litchi pericarp, their chemical composition, origin, production and so on. You cannot start the introduction directly possible probiotic effect of the aforementioned products.
7. Line 41: growing in vitro studies – please use a large number of studies. It is more academic.
8. Lines 44-45 – please rephrase, it does not have any sense.
9. Line 46 – for instead of to.
10. Line 52- polyphenols degradation and not degraded.
11. Line 52: L.plantarum in italic. Please check it in the whole manuscript. See also line 62.
12. Line 58: molecule compounds with strongly polar..this sentence is unfished
13. Lines 60-62 – please rephrase, does not have any sense.
14. Line 90. By the method ….I really believe that a native English speaker should check all the manuscript. There are a lot of grammar errors.
15. Line 91- the temperature of 80 °C is not too high for the extraction of LSPC and LPPC?
16. Why for the extraction of LPPC extract authors used 50°C and for LSPC 80°C?
17. Line 91 instead if by please use through.
18. Line 117 inoculated with four….
19. Line 135. In vitro and not in virto
20. Line 136 – the meaning of BSA under brackets. When an abbreviation it used for the first time, authors should mentioned the meaning.
21. Line 208. Specifically, instead of Specificily or you can use for instance
22. Line 210-212 – please rephrase. It has no sense written like this.
23. Line 278-279 – a sentence without a logic sense. Please rephrase it.
24. Lines 319, 358 – try not to use personal pronouns such as we
25. Please do not start the conclusion part with as discussed above. The conclusion must sum it up the main findings in an original way, so that readers could be able to understand the main results of the study, without reading the whole manuscriot.

I highly recommend that all the manuscript must be check buy a native English speaker.
Round 2
Reviewer 1 Report
Comments and Suggestions for Authors
All my comments were incorporated into the revised manuscript
Reviewer 2 Report
Comments and Suggestions for Authors
The revised manuscript seems to be appropriately modified.
Comments on the Quality of English Language"Minor revision" means that there are some typographical errors.
Reviewer 3 Report
Comments and Suggestions for Authors
I believe that the manuscript could now be published in the present form. The authors have highly improved it. Thank you!
Comments on the Quality of English LanguageIn my opinion, English language needs extra minor check. Thank you!
